# Clinical Proof-of-Concept of a Non-Gene Editing Technology Using miRNA-Based shRNA to Engineer Allogeneic CAR T-Cells

**DOI:** 10.3390/ijms26041658

**Published:** 2025-02-15

**Authors:** Caroline Lonez, Jennifer Bolsée, Fanny Huberty, Thuy Nguyen, Céline Jacques-Hespel, Sebastien Anguille, Anne Flament, Eytan Breman

**Affiliations:** 1Celyad Oncology, 1435 Mont-Saint-Guibert, Belgiumebreman@celyad.com (E.B.); 2Division of Hematology, University Hospital Antwerp (UZ Antwerp), 2650 Antwerp, Belgium

**Keywords:** CAR T-cell, allogeneic, shRNA, non-gene editing, adoptive therapy

## Abstract

With the success of chimeric antigen receptor (CAR) T-cell therapy in B-cell malignancies, efforts are being made to extend this therapy to other malignancies and broader patient populations. However, limitations associated with the time-consuming and highly personalized manufacturing of autologous CAR T-cells remain. Allogeneic CAR T-cell approaches may overcome these challenges but require further engineering to reduce their alloreactivity. As a means to prevent graft-versus-host disease (GvHD) of allogeneic CAR T-cells, we have selected a micro RNA (miRNA)-based short hairpin RNA (shRNA) targeting CD3ζ which efficiently downregulates the expression of the T-cell receptor (TCR) below detection level. We generated allogeneic anti-B-cell maturation antigen CAR T-cells (CYAD-211) that co-express an anti-CD3ζ miRNA-based shRNA within the CAR construct which efficiently inhibited TCR-mediated signaling in vitro and GvHD in vivo. CYAD-211 was subsequently evaluated in a Phase-I clinical trial (NCT04613557), in patients with relapsed or refractory multiple myeloma. No signs of GvHD were observed despite evidence of engraftment, demonstrating efficient downregulation of the TCR. Our data provide proof of concept that a non-gene-edited technology can generate fully functional allogeneic CAR T-cells, without any signs of GvHD. However, further engineering of the CAR T-cells is needed to improve their persistence and long-term activity.

## 1. Introduction

Chimeric antigen receptor (CAR) T-cell therapies showed impressive overall response rates and durable remissions in patients with B-cell malignancies such as B-cell leukemia and multiple myeloma (MM) and have paved the way for the development of CAR T-cell therapies in other indications [1,2,3].

The impressive results in heme malignancies have led to 7 FDA-approved autologous CAR T-cell therapies [4]. However, autologous CAR T-cell therapies suffer from several challenges involving (i) the patient’s own T-cells which are frequently of low quality and may even be cancerous [5,6] and (ii) a lengthy and complicated manufacturing process, that is not feasible in many hospitals or clinics, and may even come too late to be of use, as patients may relapse by the time the manufacturing is completed [7,8].

Using healthy donor cells instead of the patient’s own cells may counteract these issues, through the deliverance of a high-quality CAR T-cell therapy. However, the allogeneic T-cell receptor (TCR) on the donor T-cells may induce graft-versus-host disease (GvHD) upon recognition of the patient’s human leukocyte antigen (HLA). Therefore, allogeneic CAR T-cells need to be engineered to either have no TCR or blunted TCR signaling [9,10].

Of late, most engineering efforts to prevent GvHD have been focused on gene-editing technologies [10,11], like clustered regularly interspaced short palindromic repeats (CRISPR)-Cas9, to permanently modify the CAR T-cells at the genome level. However, editing has inherent safety concerns associated with genetic disruptions that may lead to unintended, irreversible off-target genetic alterations (i.e., off-target DNA cleavages, mutations, or chromosomal rearrangements) [12,13,14] and may also potentially lead to late differentiated or exhausted cells, with limited persistence and/or functionality [15].

An alternative approach is the use of non-gene edited methodologies such as RNA interference (RNAi), including short hairpin RNAs (shRNAs) and micro RNAs (miRNAs), which regulate gene expression post-transcriptionally, i.e., without site-directed intervention on the cell genome. shRNAs and miRNAs can be actively transcribed as precursors in the recipient cells, from a standard vector, such as a lentiviral or retroviral vector, without the need for separate promoter systems, ensuring long-lasting gene downregulation. Latest developments were focused on synthetic miRNAs, in which the guide sequence is swapped with an shRNA-based one directed against the gene of interest, exploiting the natural miRNA pathway, hence with a lower risk of cellular toxicity than shRNA [16,17]. miRNA-based approaches had already been used to successfully inhibit HIV and HCV replication [18,19,20]. However, these approaches have never been used to engineer CAR T-cells.

In this publication, we aimed to develop a non-gene edited-based technology that could render donor cells irresponsive to GvHD, by co-expressing a miRNA-based shRNA directed against the TCR subunit CD3ζ and combining it together with an all-in-one vector that contains the CAR and an additional truncated tag. At the time of this study initiation, the development of CAR T-cells in MM was expanding, following outstanding successes in B-cell malignancies and based on preliminary data obtained with CAR T-cells targeting the B-cell maturation antigen (BCMA). Indeed, BCMA is preferentially expressed on multiple myeloma plasma cells but not on hematopoietic stem cells, making it a promising antigenic target. Nevertheless, despite the approval of two BCMA CAR T-cell products in 2021 and 2022, the durability of responses still remains a therapeutic challenge, with a significant proportion of patients relapsing [21]. Retreatment with a second infusion of CAR T-cells has been considered as one option to maintain responses [22]. In this context, the development of an allogeneic BCMA CAR T-cell, readily available when a patient needs it, might have its own advantages. We therefore created the first ever non-gene-edited allogeneic BCMA targeting CAR T-cells, and evaluated this product candidate in a first-in-human Phase I clinical study where it proved to be safe and show some signs of efficacy, thus validating our approach.

## 2. Results

### 2.1. Downregulation of CD3ζ Using a miRNA-Based shRNA Efficiently Suppresses TCR-Mediated Signaling

The TCR consists of a variable TCRαβ heterodimer that is intimately associated with the non-variable signal transduction CD3 complex comprising the CD3γ, CD3δ, CD3ε and CD3ζ subunits. All TCR subunits are synthesized in excess and stored in the endoplasmic reticulum, with the exception of CD3ζ, which reaches the Golgi system independently. In the absence of CD3ζ, the remaining partial TCR complex is unable to stabilize in the cell membrane and is led to degradation [23]. We, therefore, assessed whether the targeting of CD3ζ could reduce the TCR complex expression and signaling and whether this would allow the generation of allogeneic CAR T-cells with a low risk of TCR-mediated GvHD.

Three different shRNAs targeting CD3ζ were incorporated into a miRNA scaffold within a lentiviral vector that included a truncated CD19 tag (Figure 1A).

The three miRNA-based constructs showed up to 60% reduction in CD3ζ mRNA expression levels (Figure 1B), compared to untransduced control cells, and significantly reduced both TCRα/β and CD3ε protein expression to negative levels (Figure 1C). Overall, all constructs showed very similar characteristics, although shRNA-CD3ζ #2 emerged as slightly more efficient.

To validate the chosen candidate it was assessed in primary human T-cells from 5 healthy donors. The miRNA-based construct showed significant downregulation of the CD3ζ mRNA when transduced into primary T-cells (Figure 1D) compared to control Mock transduced T-cells. A significant downregulation of both the TCRα/β and CD3 in both CD4+ and CD8+ T-cell subsets was also observed at similar levels to CRISP-Cas9 CD3ζ knockout (Figure 1E,F). The miRNA-based construct against CD3ζ negated TCR-mediated signaling, as indicated when the TCR was stimulated with anti-CD3 antibodies, similarly to a CRISPR CD3ζ control as shown in Figure 1G,H.

Off-target analysis comparing the shRNA-CD3ζ with control cells showed the expression of three genes to be significantly decreased in both CD4+ and CD8+ T-cells: CD247—the shRNA target, CD19—normally not expressed in T-cells, but expressed as a truncated form in both vectors to be used as a reporter and selection marker, and gamma-glutamyl hydrolase (GGH). Sequence analysis showed high complementarity between the shRNA sequence and GGH, making it likely a direct off-target of the shRNA-CD3ζ; however, GGH downregulation in T-cells is not predicted to result in a proliferative advantage of T-cells but rather the reverse, as GGH function is required for DNA synthesis and thus cell proliferation. The change in the CD19 reporter expression is not due to a direct effect of the shRNA but rather linked to a difference in expression efficiency between mock and shRNA-CD3ζ construct. Notably, the normalized counts of GGH and CD19 suggested a low relative abundance of the mRNA of GGH and CD19 reporter in both Mock and shRNA-transduced cells. Altogether, this indicates that the off-target profile is very limited and has no impact on the CAR T-cells (Appendix A).

### 2.2. Development of the First Non-Gene Edited Allogeneic CAR T-Cell Targeting BCMA

MM is the second most common hematological cancer and BCMA, a protein present in high concentrations on a small subset of healthy blood cells and multiple myeloma cells, is the most extensively studied CAR target for myeloma. It is relatively homogenously expressed in multiple myeloma patients making it an interesting target for allogeneic CAR T-cell therapy. Healthy-donor-derived T-cells were transduced with retroviral vectors encoding for the anti-CD3ζ miRNA-based construct, BCMA-specific CAR, and a truncated CD34 (tCD34) reporter gene (Figure 2A). Transduction efficiency (presented as the percentage of CD34+ cells) and expression of the BCMA-targeting CAR were assessed on T-cells derived from 7 healthy donors (Figure 2B). CAR efficacy was analyzed by incubating CAR T-cells for 24 h at a 1:1 ratio with the BCMA-expressing MM cancer cell line RPMI-8226 (Figure 2C,D). Cytokine secretion was highly positive when BCMA CAR T-cells were incubated with the cancer cell line, in contrast to the Mock cells (Figure 2C). Similarly, cytolytic activity, as assessed by the lactate dehydrogenase (LDH) assay (Figure 2D), showed a high release of LDH when CAR T-cells were cocultured with the cancer cells. Finally, the specificity of the BCMA-targeting CAR to the BCMA was demonstrated by the significant decrease in IFN-γ production when adding the soluble recombinant BCMA-Fc protein to counteract the BCMA-specific CAR recognition of RPMI-8226 cancer cells (Figure 2E). Additionally, a binding assay of CYAD-211 against fixed HEK293 cells expressing 5528 full-length human proteins, either plasma membrane proteins or cell surface-tethered human-secreted proteins, was performed (Appendix A). After primary and confirmation screenings, only one CAR-specific interaction was observed and corresponded to BCMA. In conclusion, no CAR-related off-target binding interactions were identified, indicating high specificity for the primary target.

To create a clinical-grade product that allows for the generation of a high number of doses from a single apheresis, the manufacturing process of the BCMA-specific CAR was optimized. A twelve (12)-day manufacturing process incorporating a transduced cell purification step (enrichment of CD34+ cells) on day 6 and TCR^+^ depletion upon harvest was selected (Figure 3A). The BCMA-specific CAR T-cells produced using this process were termed CYAD-211. CD34 cell surface expression was used as a measure of the transduction efficiency along the process (Figure 3B). Together with the selection of CD34+ cells on Day 6 and depletion of TCR+ cells on Day 12, this optimized process led to a highly pure CYAD-211 product (CD34+ and TCR−). During the process, the median number of CYAD-211 (i.e., pure transduced T-cells) produced at the end of the process per billion of peripheral blood mononuclear cells (PBMC) engaged was 8.5 billion (range 3.9–19.4 billion, Appendix A). Phenotypic analysis of the cells at the end of the process (Figure 3C,D) showed that more than 99.8% of CYAD-211 were negative for TCRα/β cell surface expression (TCR−), illustrating the efficient downregulation of TCR and the effective elimination of any remaining non-transduced cells at the end of the process.

Additionally, we demonstrated that the CYAD-211 cells produced from three different donors were able to secrete IFN-γ upon co-culture with BCMA-expressing tumor cells (Appendix A). More detailed analysis of the CYAD-211 phenotype (Appendix A–G) from three different donors showed that the CD4/CD8 ratio was donor-dependent, with the majority of cells being non-activated (CD25−/CD69−, 68 ± 18%), or non-exhausted (PD-1−/Lag3−, 95 ± 1.5%) and displaying an early differentiated profile (CD45RA−/CD62L+ and CD45RA+/CD62L+, 65 ± 14%) at the end of the process. Additionally, all CYAD-211 cell batches displayed a vector copy number (VCN) below 5 copies per cell (median: 2.47, range: 2.38–3.39) and no detectable replication-competent retrovirus (RCR) in any of the samples, confirming compliance to regulatory requirements for clinical evaluation.

### 2.3. CYAD-211 Displays No GvHD, and Enhanced Efficacy Against a Highly Potent In Vivo MM Mouse Model

CYAD-211 was evaluated in vivo in a gold-standard preclinical model of GvHD. As shown in Figure 4A, control (Mock) T-cells induced GvHD, manifested with weight loss onset as of day 23 and a median survival of 32 days. In contrast, limited and transient weight loss was observed in mice who were injected with CYAD-211 derived from the same donor. After initial engraftment similar to the control T-cells, CYAD-211 cell frequency remained relatively stable until day 14 and decreased thereafter to reach undetectable levels by day 42 (Figure 4C). Importantly, all mice injected with CYAD-211, from any of the 3 donors, survived until the end of the experiment on Day 69 (Figure 4B) demonstrating that intravenous injection of CYAD-211 was well tolerated, did not produce any significant systemic toxicity, and showed no evidence of xenogeneic GvHD.

Finally, we investigated the in vivo anti-tumor activity of CYAD-211 in a xenograft model of MM using the KMS-11 cell line. Whilst mice that received vehicle injections or control (Mock) T-cells succumbed to the tumor with a median survival of 35 and 43 days, respectively, mice injected with CYAD-211 (from two different donors) remained alive until the end of the experiment at Day 63 (Figure 4D). Bioluminescence kinetics emitted by luciferase-expressing KMS11-luc tumor cells is shown in Figure 4E, with images of individual mice at different timepoints shown in Figure 4G. All mice injected with the vehicle showed a rapid progression of their tumor burden and had to be sacrificed between Day 35 and Day 37 due to hind limb paralysis and body weight loss. Mice treated with Mock T-cells had a slightly delayed tumor progression. In contrast, to CYAD-211 treated mice, the onset of increase in the bioluminescence values was delayed by at least 3 weeks. Engraftment of the CYAD-211 in the peripheral blood is shown in Figure 4F. Similar data were obtained with another MM xenograft model using the RPMI-8226 cell line (Appendix A).

### 2.4. Clinical Evaluation of CYAD-211 Provides Proof-of-Concept of the miRNA-Based Construct Against CD3ζ to Control GvHD in Multiple Myeloma Patients

CYAD-211 was evaluated in the open-label Phase 1 IMMUNICY-1 trial (NCT04613557, EudraCT 2020-001414-38) in adult patients with relapsed or refractory (r/r) MM following at least two prior regimens.

During the dose-escalation segment, patients received non-myeloablative preconditioning (cyclophosphamide 300 mg/m^2^/day and fludarabine 30 mg/m^2^/day, for 3 days) followed by a single CYAD-211 infusion at three different dose-levels (DL): 3 × 10^7^, 1 × 10^8^ and 3 × 10^8^ cells/infusion.

Between 24 November 2020 and 16 August 2021, in total, twelve patients were enrolled across the 3 DLs in the dose escalation segment. Patients’ main baseline characteristics are shown in Table 1. Overall, enrolled patients were heavily pretreated patients with a median of four prior lines of treatment and more than half of the patients had high-risk MM.

Table 2 shows the safety profile of CYAD-211. Overall, CYAD-211 was well tolerated with no evidence of GvHD, dose-limiting toxicity (DLT), nor CAR-T-cell-related encephalopathy syndrome (CRES) at the 3 dose levels and the maximum tolerated dose was not reached. Globally, all patients experienced at least one adverse event (AE) of any grade. Ten patients experienced AEs related to CYAD-211 including five patients who experienced treatment-related AEs of grade 3 or above. Main CYAD-211 treatment-related AEs were abnormal laboratory test results (including decreased white blood cell or lymphocyte counts, observed in 66.7% of patients), blood and lymphatic system disorders (like neutropenia and thrombocytopenia, observed in 16.7% of patients), gastrointestinal disorders (16.7% of patients) or headaches (16.7% of patients). One patient at DL1 developed grade 1 cytokine release syndrome (CRS) requiring hospitalization which occurred 10 days post-CYAD-211 administration and was considered a serious AE (SAE). The second CYAD-211-related SAE was a herpes virus simplex (HSV) encephalitis grade 5 which occurred 10 months post CYAD-211 infusion in a patient at DL2 and was considered unrelated to CYAD-211 by the Sponsor. Infections were observed in 5 patients, with only one considered as related to CYAD-211 treatment (in the patient presenting HSV encephalitis).

Response to CYAD-211 and duration of response at the three dose levels are shown in Figure 5A. Three patients achieved partial response per International Myeloma Working Group (IMWG) criteria while 9 patients had stable disease including one patient with evidence of a reduction in the size of soft tissue plasmacytomas in some lesions (Appendix A).

Analysis of peripheral blood samples confirmed the engraftment of CYAD-211 at all dose levels (Figure 5B). However, the engraftment was short lasting (3–4 weeks). By comparing the CYAD-211 peripheral blood kinetics and the white blood cell recovery after lymphodepletion (Figure 5B,C), we were able to demonstrate a correlation between the depth of lymphodepletion and engraftment over time (Appendix A). To note, none of the patients presented anti-donor antibodies two months post CYAD-211 infusion despite multiple mismatches with the donor’s HLA genotyping (Appendix A).

The levels of systemic cytokines and chemokines in serum samples obtained before treatment and at the peak of engraftment were analyzed by Luminex profile (Appendix A). Interestingly, a diminution of the proportion of stimulatory cytokines was observed in most of the patients, together with an increase in chemoattracting chemokines, suggesting that the infused CAR T-cells may modulate cytokine profile at their peak of expansion. These results combined indicate that while CYAD-211 was well tolerated with evidence of anti-tumor activity, the cells were relatively quickly rejected by the resurgence of the host immune system, likely limiting their activity and durability of response.

## 3. Discussion

This last decade has seen the emergence of autologous CAR-Ts as one of the major breakthroughs in cancer immunotherapy, with no less than 7 approved products by the FDA since 2017. However, to extend the accessibility of this therapy to a broader patient population, the development of allogeneic “off-the-shelf” approaches is strongly needed. To achieve this, multiple gene editing technologies are used to abrogate the expression of endogenous TCRs in order to minimize the risk of GvHD [26]. In addition, the possibility of targeted integration allows for the simultaneous inactivation of the targeted gene while inserting the CAR construct, which avoids the oncogenic risks associated with semi-random integration of viral vectors. To date, several gene-edited allogeneic CAR T-cell candidates have shown clinical activity, together with their demonstrated ability to abrogate the risk of GvHD, in clinic [27]. Nevertheless, the use of gene-editing technologies (such as CRISPR/Cas9) leads to double-stranded breaks, disruption of the normal genomic sequence, or chromosomal abnormalities with unpredictable consequences which are a major safety concern [27]. Furthermore, the irreversibility of such technologies makes off-target effects an important concern (see [27,28,29] for reviews on these aspects). We therefore aimed to develop an alternative, miRNA-based shRNA technology for the generation of allogeneic CAR-Ts, and more generally, as a platform to engineer CAR T-cells to further endow them with specific properties.

We present here for the first time the use of RNAi as a means to reduce the TCR expression at the CAR T-cell surface, impairing their ability to induce GvHD, similar to what has been achieved with gene editing methods. Our screening identified shRNA-derived sequences capable of knocking down the CD3ζ TCR subunit in T-cell lines and primary T-cells, effectively impairing their ability to induce IFN-γ release upon TCR stimulation. Importantly, the reduction in the TCR signaling in cells transduced with the miRNA-based construct-containing vector was similar to the CD3ζ knock-out cells engineered via CRISPR/Cas9 gene editing. The simplicity of using a single vector for the transduction of the primary T-cells for the creation of CYAD-211 provides several additional benefits in comparison to gene editing technologies, where multiple rounds of transductions and subsequent purifications are needed to achieve a pure population. Interestingly, through a simple process of enrichment and depletion, we could achieve a very high yield and a highly pure population adapted for clinical use.

Clinical evaluation of CYAD-211 in patients with MM confirmed the good safety profile of CYAD-211, with no evidence of GvHD observed in any of the 12 patients evaluated. However, it is clear that to improve the anti-tumor efficacy and cellular persistence of the CAR used in this trial, additional elements need to be considered. The lack of observed GvHD despite engraftment of CYAD-211 provides proof of concept of the safe administration of CAR T-cells using a miRNA-based shRNA allogeneic platform and validates the use of non-gene editing technologies for the generation of allogeneic CAR T-cells.

The short duration of cells post-infusion might be linked to the specificity of the indication as the lack of CAR T-cell persistence likely related to their exhaustion is known to contribute to recurrence in MM patients [30]. The lack of sustained engraftment of CYAD-211 is also most likely due to the rejection of the allogeneic cells by the recovering immune system of the recipient (via host-versus-graft [HvG] reaction). These results are comparable to the lack of persistence observed in other clinical trials with allogeneic CAR T-cells engineered with gene-editing technologies and call for additional engineering to prevent HvG reaction [27]. Several strategies to limit the HvG reaction are currently under evaluation including knocking out the β2-microglobulin subunit of HLA-I molecules, CD47 overexpression, or removal of CD52 but no strategy seems to be sufficient on its own to fully avoid the CAR T-cell allorejection [27]. Hence, it is becoming clear that the future of allogeneic CAR-T therapies would require diverse modifications to downregulate or disrupt genes involved in allorejection, in addition to those targeted for the prevention of GvHD. Through multiplexing of the miRNA scaffold to include multiple additional targets within a single CAR vector, we may knock down other molecules that are important in driving immune rejection (e.g., major histocompatibility complex class [MHC] I or II) or those involved in the immunosuppressive tumor microenvironment (e.g., PD-1, CTLA-4, LAG-3, TIM-3, and TIGIT) that also contribute to inhibit T-cell proliferation and activity, thus allowing for enhanced persistence [31]. Other groups have shown pre-clinically that gene-editing of MHC-I, II as well as other genes can lead to enhanced persistence [32]. However, this has not yet been shown clinically.

Overall, our data provide proof of the principle that miRNA-based shRNA-mediated knock-down can generate fully functional ‘off-the-shelf’ allogeneic CAR T-cells without any signs of GvHD in humans.

## 4. Materials and Methods

### 4.1. Plasmid Construction and Retroviral Vector Production

Retroviral vectors were generated using the pSFG backbone. BCMA CAR was generated using the BCMA-specific scFv (sequence provided in Appendix A) fused to hinge (55 aa) and transmembrane domains from CD8α, connected to the intracellular, signaling domains of tumor necrosis factor receptor superfamily member 9 (alternatively termed 4-1-BB) and of the TCR component CD3ζ. The truncated form of CD34 (tCD34) reporter protein consisting of amino acids 1 to 328 of the human CD34 protein was included within the transgene cassette to enable identification and purification of transduced cells. miRNA-based shRNA for human CD247 (based on the miR-196a-2 scaffold) was polymerase chain reaction (PCR) amplified from the SMARTvector backbone (see reference below under Lentiviral vector production) before being ligated into the pSFG plasmid, using XhoI and Kpn2I restriction sites (ThermoFisher Scientific, Waltham, MA, USA). Retroviral vectors were then generated in two steps, as described in detail elsewhere [31]. Briefly, transfection of a certified ecotropic retroviral packaging cell line (Phoenix-eco cells (ATCC^®^, Manassas, VA, USA)) with the plasmid of interest was performed using calcium phosphate. Vector supernatant was used to transduce a certified packaging cell line, PG-13 (ATCC^®^), developed by Miller et al. [33], to obtain the retroviral vector-producing PG-13 cell line. Stable PG13 producer cells were seeded, and the final retroviral vector was collected 48 and 72 h post seeding.

### 4.2. CAR T-Cell Production

For clinical grade CAR T-cell production, PBMCs from healthy donors were isolated from aphaeretic material (HemaCare, Northridge, CA, USA) and activated for 48 h in X-Vivo 15 medium (Lonza, Basel, Switzerland) supplemented with IL-2 (100 IU/mL) and Transact (Miltenyi Biotec, Bergisch Gladbach, Germany) as detailed in [31]. Cells were subsequently harvested and transduced with a retroviral vector for 48 h in the presence of IL-2 and Akti-1/2 (Bio-Techne, Minneapolis, MN, USA). The vector titer was adjusted to ensure a VCN below 5. On day 4 post activation, transduced cells were expanded for a further 48 h in the presence of IL-2 and Akti-1/2. On day 6 post activation, transduced cells were enriched using the CD34 MicroBead Kit (Miltenyi) with a CliniMACS system. Selected cells were further expanded in complete X-Vivo 15 medium in the presence of IL-2 and Akti-1/2 and harvested at day 10. Finally, the remaining TCRα/β positive cells were depleted using a TCR MicroBead Depletion Kit (Miltenyi) with a CliniMACS system and the final product was cryopreserved.

### 4.3. Lentiviral Vector Production

The lentiviral plasmids used for shRNA selection were obtained from Horizon Discovery/Dharmacon: shRNA-CD3ζ-1 (Cat. 7270125), shRNA-CD3ζ-2 (Cat. 8938341), shRNA-CD3ζ-3 (Cat. 5172810), shRNA-Neg (Cat. 29189570). The lentiviral production was performed using the Lentiviral Packaging Plasmid mix from Cellecta following the manufacturer’s instructions.

Jurkat T-cells, clone E6-1 (ATCC^®^) were maintained in RPMI medium (Lonza), supplemented with 10% FBS, 1% GlutaMAX (Invitrogen, Waltham, MA USA), and 1% Pen-Strep. Jurkat cells as well as human primary T-cells were activated for 48h in the presence of CD3 antibodies. On Day 2, cells were expanded for a further 48h in the presence of IL-2 and transduced in retronectin-coated wells. Two days later Puromycin selection (1 µg/mL) was applied until the harvest of the cells.

### 4.4. CRISPRs

The crRNA-CD3ζ (TGGCCCTGCTGGTACGCGG) was custom-designed at IDT. crRNA-CD3ζ and Alt-R^®^ CRISPR-Cas9 tracrRNA oligos were mixed in equimolar concentrations in a sterile microcentrifuge tube to a final duplex concentration of 100 µM, heated at 95 °C for 5 min, and cooled to room temperature (15–25 °C). The ribonucleoprotein (RNP) complex, obtained by diluting the crRNA-tracrRNA duplex and Alt-R^®^ S.p. Cas9 Nuclease V3 components in PBS, was then incubated at room temperature for 10–20 min before being electroporated in cells with the 4D-Nucleofector™ System (Lonza).

### 4.5. RNA Extraction and qPCR

Total RNA was extracted using the RNeasy mini kit (Qiagen, Hilden, Germany) following the manufacturer’s protocol. Reverse-transcription and gene expression analysis were performed in a one-step real-time qPCR by means of the LightCycler 480 RNA Master Hydrolysis Probes (Roche, Basel, Switzerland) on a LightCycler LC480 instrument (Roche). The following primers were used for detection: CD247 set (ThermoFisher, Hs00609515_m1), CD3E set (ThermoFisher, Hs01062241_m1).

### 4.6. Immunophenotyping by Flow Cytometry

Flow cytometry analysis was performed at several steps along the process. In brief, cells were harvested, washed with Attune Focusing Fluid, and resuspended in 50 µL Attune Focusing Fluid. 50 µL of the antibody mix for the indicated Panel was added, followed by 30 min incubation time. Subsequently, 900 µL Attune Focusing Fluid was added and cells were acquired on an Attune NxT Flow Cytometer, ThermoFisher Scientific, Waltham, MA, USA). Cell surface staining was performed using the following fluorochrome-labeled antibodies: recombinant human BCMA/TNFRSF17 Fc (R&D Systems, Minneapolis, MN, USA), TCRα/β-APC (Biolegend, San Diego, CA, USA) CD3-BV711 (BD), CD4-APC-H7 (BD), Live/Dead Fixable Aqua (ThermoFisher), CD8-BV605 (Biolegend), CD69-APC (BD), CD279-PE (ThermoFisher), CD223-APC-R700 (BD), CD25-AF488 (Thermo), CD62L-PE-Cy7 (eBioscience, San Diego, CA, USA), CD45RA-APC (ThermoFisher Scientific).

### 4.7. TCR Activation Assay

1 × 10^5^ T-cells were seeded per well in a 96-well plate. T-cells were activated for 24 h with different concentrations (from 2 to 200 ng/mL) of anti-CD3 antibody (clone OKT3). After 24 h supernatant was collected and IFN-γ concentration was measured by ELISA using a Human IFN-γ Quantikine ELISA Kit (R&D Systems) according to manufacturer’s instructions.

### 4.8. In Vivo Studies

To assess potential GvHD induction, NOD SCID gamma-c^−/−^ (Non-Obese Diabetic Severe Combined ImmunoDeficiency Gamma—NSG) mice were intravenously (IV) injected with 20 × 10^6^ CYAD-211 CAR T-cells, 20 × 10^6^ Mock T-cells (as a positive control of GvHD induction) or vehicle (as a negative control of GvHD induction). Appearance of GvHD signs was assessed by monitoring indicative clinical signs, such as fur texture, skin, posture, breathing, activity, and body weight, three times a week until the end of the study.

To assess anti-tumor efficacy, NSG mice were intravenously injected with 5 × 10^6^ KMS-11 luciferase cells on day 0. On day 5, animals were distributed into experimental groups, taking into consideration individual bioluminescence data in order to have groups homogeneous for the average of radiance values (photon/second/cm^2^/steradian). In addition, body weight, general condition, and aggressiveness were considered during the group/cage allocation procedure. On day 6, mice were intravenously injected with 10 × 10^6^ BCMA CAR-shRNA CD247 CAR T-cells, 10 × 10^6^ Mock T-cells, or a vehicle. Anti-tumor activity was evaluated every two weeks by measuring tumor cell growth by in vivo imaging. In addition, mice were monitored daily for mortality and clinical signs, and body weights were measured twice a week.

### 4.9. Clinical Trial Design and Participants

IMMUNICY-1 (CYAD-211-001) is a phase I, open-label trial conducted at three sites in Belgium (EudraCT 2020-001414-38) and two sites in the United States of America (NCT04613557) (see Appendix B). The dose-escalation segment evaluated three DLs of CYAD-211 (DL1: 3 × 10^7^, DL2: 1 × 10^8,^ and DL3: 3 × 10^8^ total cells per each infusion, to be adjusted per body weight for patients weighing ≤65 kg at 4.6 × 10^5^ CYAD-211/kg, 1.5 × 10^6^ CYAD-211/kg, and 4.6 × 10^6^ CYAD-211/kg, respectively) using a Fibonacci 3+3 design to determine the recommended dose of CYAD-211 when administered as a single infusion after a non-myeloablative preconditioning chemotherapy. Patients received a preconditioning chemotherapy regimen (cyclophosphamide 300 mg/m^2^/day and fludarabine 30 mg/m^2^/day, for 3 days) followed by a single infusion of CYAD 211 infusion on Day (D) 1.

Patients were 18 years of age and older with an Eastern Cooperative Oncology Group (ECOG) performance status ≤2. Eligible patients had documented diagnosis of MM r/r to at least two prior MM treatment regimens which should include exposure to immunomodulatory drugs (e.g., lenalidomide, pomalidomide) and proteasome inhibitors (e.g., bortezomib, carfilzomib, ixazomib) either alone or in combination. Prior treatment with daratumumab, elotuzumab, or selinexor was allowed. Patients must have had a response (minimal or better) to at least one prior regimen. Presence of measurable disease as per IMWG Response Criteria [34], was defined as one or more of the following (a) serum myeloma (M) protein ≥ 0.5 g/dL, (b) urine M protein ≥ 200 mg/24 h, (c) serum immunoglobulin (Ig) free light chain (FLC) ≥ 10 mg/dL with abnormal serum Ig kappa-lambda FLC ratio, and (d) imaging consistent plasmacytoma with the presence of any clonal plasma cells in peripheral blood or bone marrow. Additional baseline assessments required for eligibility, as assessed by standard laboratory criteria and spirometry, included adequate hepatic and renal functions, a left ventricular ejection fraction of ≥40%, and a forced expiratory volume in the first second (FEV-1)/forced vital capacity (FVC) ≥ 0.7 with FEV-1 ≥ 50% predicted. Serious uncontrolled medical disorders, persistent toxicities greater than or equal to Common Terminology Criteria for Adverse Events (CTCAE) grade 2 caused by previous cancer therapy, active infections, or autologous stem cell transplant within 12 weeks of registration or an allogeneic stem cell transplant within 6 months of starting study treatment were the main exclusion criteria.

### 4.10. Clinical Trial Endpoints and Procedures

The primary endpoint of this open-label phase I dose escalation study was the occurrence of DLTs at any time after initiation of the CYAD-211 infusion up to D36. DLT was defined as a toxicity at least possibly related to the CYAD-211 treatment including (1) any CTCAE or CRS/CRES grade 5 toxicity not due to the underlying malignancy, (2) any CTCAE Grade 3 or higher allergic reactions related to the CYAD 211 infusion that cannot be controlled to ≤Grade 1 within 24 h of treatment with appropriate treatment, (3) any CTCAE Grade 4 fever that cannot be controlled to ≤Grade 3 with antipyretics treatment, (4) any CTCAE Grade 3 or higher CYAD-211-related toxicity that could lead to irreversible damage of a vital organ, (5) any CTCAE Grade 4 CYAD 211-related neutropenia or thrombocytopenia not associated with disseminated intravascular coagulation that cannot be controlled to ≤Grade 2 or to patient baseline hematological values within 42 days with appropriate treatment, (6) any Grade 4 CRS of any duration or any Grade 3 CRS that affects cardiovascular or pulmonary function, causes any CRS-related Grade 3 toxicity to other organs with the exception of liver and kidneys and lasting more than 72 h, or causes any CRS-related Grade 3 renal or hepatic toxicity lasting more than 7 days, (7) any Grade 3 or higher CRES of any duration, (8) any Grade 4 acute GvHD of any duration or any Grade 3 or higher acute GvHD that cannot be controlled to ≤Grade 2 within 7 days with appropriate treatment, (9) severe chronic GvHD of any duration or any moderate GVHD that cannot be controlled within 7 days with appropriate treatment.

Disease assessment in the peripheral blood (PB), in the BM or urine was performed at baseline, then on D22, D36, Month (M) 2, M3, M4, M6, M9, M12, M15, M18, M21, and M24 after the CYAD-211 infusion. Responses were assessed by the investigators according to the IMWG Response Criteria [34]. The severity of AEs was graded according to the National Cancer Institute (NCI)’s CTCAE (version 5.0) and evaluated at each visit. The severity of CRS and CRES was graded by the guidelines from the CAR-T-cell therapy-associated TOXicity (CARTOX) working group [25].

All statistical analyses for this open-label phase I dose escalation study were primarily descriptive. Data were analyzed with SAS version 9.4 software. For biomarkers and cellular kinetics, GraphPad Prism software version 10.4.1 was used.

## 5. Patents

The following patents and patent applications are associated with this work: US20220202863 and US20120321667.

## Figures and Tables

**Figure 1 ijms-26-01658-f001:**
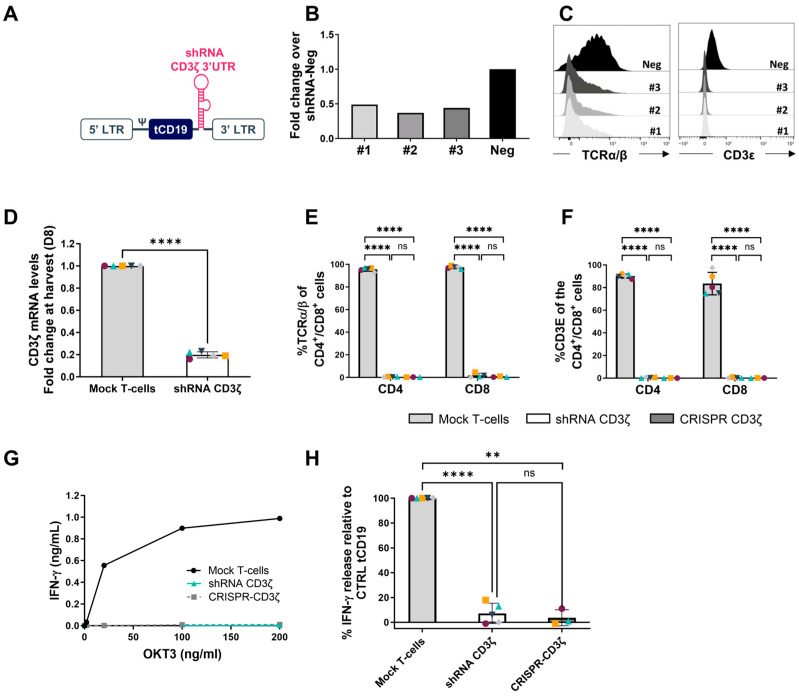
Screening of shRNAs against CD3ζ. (**A**) Schematic representation of the miRNA-based CD3ζ shRNA genetic constructs. (**B**) Screening of different (#1, #2, and #3) shRNAs targeting the 3′ untranslated region (UTR) of CD3ζ and their impact on mRNA levels of CD3ζ after transduction in Jurkat T-cells using lentiviral vectors, compared to untransduced control cells, evaluated by qPCR at harvest. (**C**) Representative flow cytometry histograms illustrating surface expression of TCRα/β and CD3ε in Jurkat cells transduced with control vector (Neg) or vector containing miRNA-based construct targeting CD3ζ. Cells were gated on SSC/FSC for lymphocytes and the overlaid histograms of TCRα/β+ (left) and CD3ε+ (right) within the viable cells are displayed. (**D**) Impact of selected miRNA-based construct (#2) transduction on CD3ζ mRNA levels in primary T-cells. Data for each of the T-cell arms are presented as mean +/− SD. Each individual point represents a donor. **** *p* value of <0.0001. (**E**–**H**) Impact of selected miRNA-based construct (#2) transduction in primary T-cells from healthy donors, and as compared to primary T-cells transiently nucleofected with ribonucleoproteins of Cas9 protein assembled with tracrRNA-crRNA-CD247 duplex (CRISPR CD3ζ). (**E**,**F**) Histograms illustrating the % of TCRα/β^+^ and CD3ε^+^ cells in CD4+ and CD8+ T-cells of each arm at harvest (day 8) as evaluated by flow cytometry. Bar graph data are presented as mean +/− SD. Each individual point represents a donor. **** *p* value of <0.0001. ns: not significant. (**G**) T-cells (from one representative healthy donor) transduced with a vector containing miRNA-based construct #2 targeting CD3ζ or with control vector (Mock T-cells) or T-cells nucleofected with CRISPR against CD3ζ, were stimulated through TCR using anti-CD3 antibody (OKT3). After a period of incubation of 24 h, the IFN-γ secretion in the supernatant (ng/mL) was quantified by ELISA as a readout for T-cell activation. (**H**) Percentage of IFN-γ secretion in response to 200 ng/mL OKT3 stimulation per T-cell arm. Data are presented as mean +/− SD. Each individual point represents a donor. **** *p* value of <0.0001. ** *p* value of <0.01. ns: not significant.

**Figure 2 ijms-26-01658-f002:**
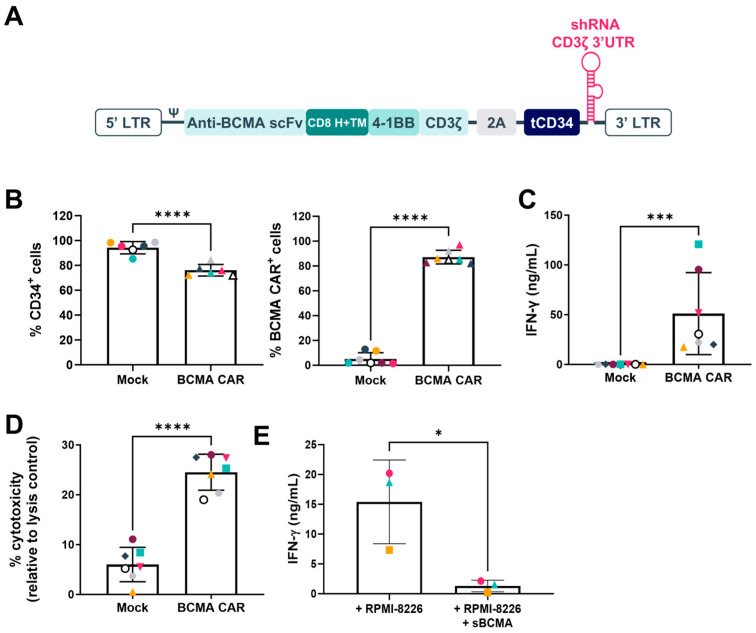
Functionality of BCMA-specific CAR expressing shRNA CD3ζ. (**A**) Schematic representation of BCMA-specific CAR-shRNA CD3ζ genetic constructs. CAR, chimeric antigen receptor; LTR, long terminal repeat sequence; ψ, retroviral Psi packaging element; 2A, 2A self-cleaving peptides; shRNA, short hairpin ribonucleic acid; tCD34, truncated CD34; UTR, untranslated regions. (**B**) Percentage of transduced CD34+ cells (left) or BCMA CAR+ cells (right) among living cells at harvest (day 10) of CAR T-cells generated from 7 healthy donors measured by flow cytometry. (**C**) IFN-γ secretion in the supernatant upon 24 h co-culture of CAR T-cells generated from 7 healthy donors with multiple myeloma cancer cell line RPMI-8226 (1:1 ratio), as measured by ELISA. (**D**) Cytotoxic activity was measured using an LDH assay in the supernatant of 24 h co-cultures of CAR T-cells generated from 7 healthy donors with multiple myeloma cancer cell line RPMI-8226 (1:1 ratio). (**E**) IFN-γ release of BCMA-specific CAR generated from 3 healthy donors in an RPMI-8226 co-culture with or without soluble recombinant BCMA-Fc protein (R&D Systems, 5 µg/mL). Represented are the mean ± SD. Each symbol represents one donor. **** adjusted *p* value of <0.0001. *** adjusted *p* value of <0.001. * adjusted *p* value of <0.1.

**Figure 3 ijms-26-01658-f003:**
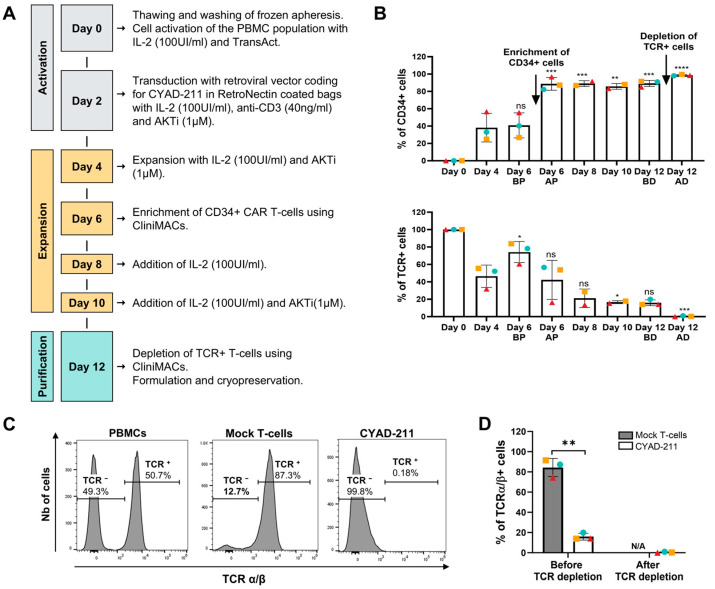
CYAD-211 clinical product development and identity. (**A**) Process flow chart of the CYAD-211 clinical product. (**B**) Percentage of transduced cells (CD34 positive population, upper panel) or TCR positive cells (TCR+, lower panel) over time during the manufacturing process. BP: before CD34+ purification step, AP: after CD34+ purification step; BD: before TCR+ depletion, AD: after TCR+ depletion. **** *p* value of <0.0001, *** *p* value of <0.001, ** *p* value of <0.01, * *p* value of <0.1, ns: not significant (versus Day 4). (**C**) Representative flow cytometry histogram plots of TCR expression of control T-cells transfected with a vector expressing only the tCD34 reporter (Mock) and CYAD-211 cells at Day 12. (**D**) Percentage of T-cell receptor positive (TCR+) cells in Mock T-cells or CYAD-211 before and after depletion of TCR positive cells at Day 12. ** *p* value of <0.01.

**Figure 4 ijms-26-01658-f004:**
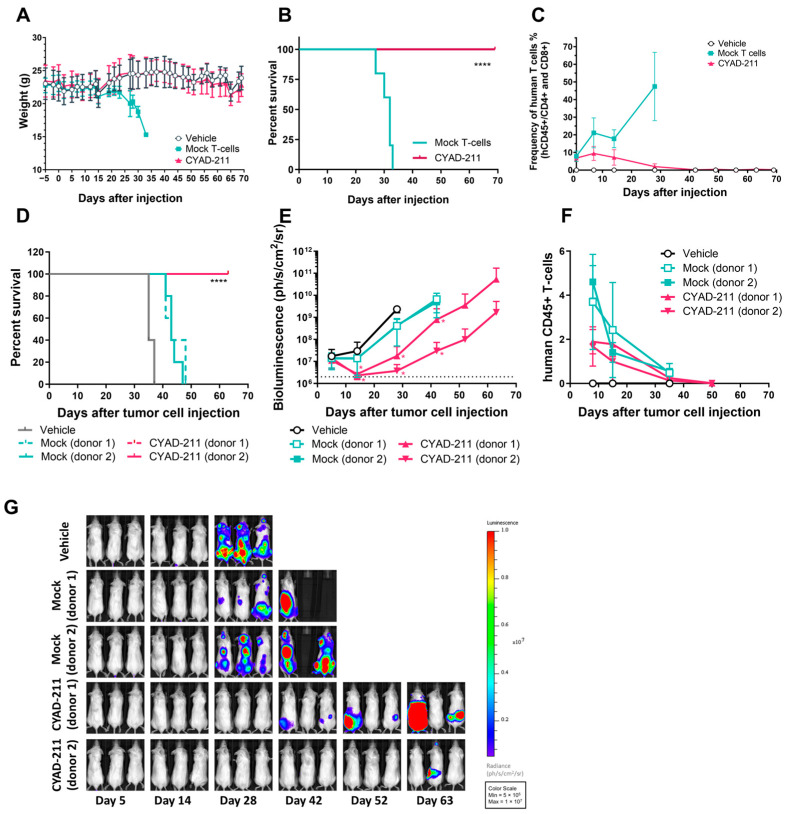
In vivo evidence of CYAD-211 lack of alloreactivity and anti-tumor activity. (**A**–**C**) Immunodeficient NSG mice (n = 5 per group) received 1.44 Gy total body irradiation one day prior to a single intravenous injection of vehicle or 20 × 10^6^, control T-cells (Mock) or CYAD-211 (cells from 1 representative donor out of three) as a model of in vivo GvHD [24]. Weight kinetics as mean percentages ± SD. **** *p* value of <0.0001, (**B**) or Kaplan–Meier survival curves of mice injected with cells from one representative, and (**C**) human T-cell engraftment in the mouse peripheral blood as mean percentages ± SD as assessed by flow cytometry (hCD45+ hCD4+/hCD8+ viable cells). (**D**–**G**) NSG mice (n = 5 per group) were injected intravenously with the vehicle, 10^7^ CYAD-211 or 10^7^ T-cells transduced with control vector (Mock), 6 days following intravenous injection of 5 × 10^6^ KMS-11-luc multiple myeloma cancer cells. (**D**) Kaplan–Meier survival curves, (**E**) kinetics of bioluminescence emitted by luciferase-expressing KMS11-luc tumor cells, (**F**) human T-cell engraftment in the mouse peripheral blood as assessed by flow cytometry (hCD45+ hCD4+/hCD8+ viable cells) and (**G**) bioluminescence images of individual mice. **** *p* value of <0.0001, * *p* value of <0.1 (versus Mock).

**Figure 5 ijms-26-01658-f005:**
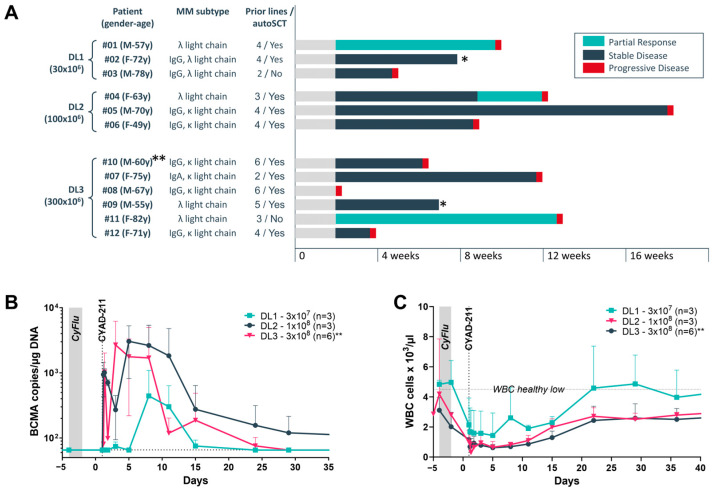
Clinical evaluation of CYAD-211. (**A**) Response and duration of response until progression in IMMUNICY-1. (**B**) CYAD-211 cell kinetics were determined by digital droplet polymerase chain reaction on genomic DNA from peripheral blood mononuclear cells isolated from blood collected at pre-specified timepoints. Values below the lower limit of quantification (LLOQ) were set at LLOQ (dotted line, 65.2 copies/µg DNA). Data were pooled per dose level (DL). * Patient discontinuation prior to disease progression; ** one patient was enrolled in Cohort 3 but due to a human error while preparing the CYAD-211 infusion, the patient received a third of a dose-level 3, corresponding to a dose-level 2. (**C**) Absolute white blood cells (WBC) kinetics in the peripheral blood. Value considered as a healthy minimal level is indicated in the dotted line (dotted line, 4.5 × 10^3^ cells/µL). Data were pooled per dose level, as in (**B**). CyFlu: cyclophosphamide and fludarabine preconditioning.

**Table 1 ijms-26-01658-t001:** Patients’ main characteristics per cohort in IMMUNICY-1 study.

	DL1 (N = 3)	DL2 (N = 3)	DL3 (N = 6) ^d^	Total (N = 12)
Age, median (range), (years)	72 (57–78)	63 (49–70)	69 (55–82)	68.5 (49–82)
Gender, %, (Male/Female)	67/33	33/67	50/50	50/50
ECOG score at screening, %, 0/1	67/33	67/33	83/17	75/25
R-ISS Stage at screening, %, 1/2/3	0/67/33	0/67/33	17/33/50	8/50/42
mSMART risk: High ^1^ (%)	50 ^a^	67	67	64 ^b^
Extramedullary disease (%)	0	33	33	25
Time between diagnosis and CYAD-211 infusion, median (range), (years) ^c^	6.8 (6.4–7.5)	7 (3.1–7.3)	6 (2.6–22.2)	6.9 (2.6–22.2)
Prior lines of therapy, median (range), (number)	4 (2–4)	4 (3–4)	4.5 (2–6)	4 (2–6)
Prior autologous SCT (%)	67	100	83	83
At least triple-exposed (%)	67	100	83	83

^1^ High-risk genetic abnormalities according to mSMART 3.0 are t(4;14), t(14;16), t(14;20), del 17p, p53 mutation and gain 1q; ^a^ 2 out of 3 patients are High and Standard Risk. The last one: is unknown due to non-contributary analysis on bone marrow (BM); ^b^ 7 out of 12 patients are high risk. The last one has unknown risk due to non-contributary analysis on BM; ^c^ incomplete dates of diagnosis were replaced by the first day of the month of diagnosis. ^d^ One patient was enrolled in Cohort 3 but due to a human error while preparing the CYAD-211 infusion, the patient received a third of a dose level 3, corresponding to a dose level 2. ECOG: Eastern Cooperative Oncology Group; ISS: International Staging System; SCT: stem cell transplantation.

**Table 2 ijms-26-01658-t002:** Main treatment adverse events per grade reported in the IMMUNICY-1 study per cohort.

	DL1 (N = 3)		DL2 (N = 3)		DL3 (N = 6)		Total (N = 12)		
	Grade								
	**<3**	**≥3**	**<3**	**≥3**	**<3**	**≥3**	**<3**	**≥3**	**All**
All AE	3 (25%)	1 (8%)	3 (25%)	3 (25%)	5 (42%)	6 (50%)	11 (92%)	10 (83%)	12 (100%)
AE related to CYAD-211 ^1^	2 (17%)	-	3 (25%)	2 (17%)	4 (33%)	3 (25%)	9 (75%)	5 (42%)	10 (83%)
SAE related to CYAD-211 ^1^	1 (8%)	-	-	1 (8%) *	-	-			2 (17%)
CRS ^2^	1 (8%)	-	-	-	-	-	-	-	1 (8%)
CRES ^2^	-	-	-	-	-	-	-	-	-
GvHD	-	-	-	-	-	-	-	-	-
Infection ^3^	-	-	2 (17%)	1 (8%) *	1 (8%)	1 (8%)	3 (25%)	2 (17%)	5 (42%)
Infusion reaction to CYAD-211	-	-	-	-	-	-			-

^1.^ Related adverse events are defined as definitely, possibly, or probably related to the CYAD-211 treatment. ^2.^ As per CAR-T-cell-therapy-associated TOXicity (CARTOX) Working Group criteria from Neelapu et al. [25]. ^3^ Including bacterial, viral, and fungal infections. * One patient had a reported grade 5 Herpes Simplex Virus [HSV] encephalitis during the treatment follow-up period considered as possibly related to CYAD-211 by the investigator but the sponsor considered the event of death to be unlikely related to CYAD-211 infusion due to the autopsy result, the low biological plausibility, the negativity of the batch infused to the patient for HSV and the 10-month time to onset between the CAR T-cell infusion and the patient’s death. AE: adverse event; CRES: CAR-T cell-related encephalopathy syndrome; CRS: Cytokine Release Syndrome; GvHD: Graft versus host disease; SAE: serious adverse event.

## Data Availability

Upon email request (contactus@celyad.com), preclinical datasets or individual participant de-identified datasets will be made available on a case-by-case basis. Supporting documents that will be available upon request include the study protocol and the informed consent form.

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
