# Peer review of "Clinical Proof-of-Concept of a Non-Gene Editing Technology Using miRNA-Based shRNA to Engineer Allogeneic CAR T-Cells"

_ijms, 2025, doi:10.3390/ijms26041658_

Round 1

Reviewer 1 Report

Comments and Suggestions for Authors
  1. Please add the BCMA expression of the cell line-RPMI 8226
  2. Any other cell lines tested for cytotoxicity in vitro, any other E:T ratios?
  3. In figure 2E, the IFN-g levels were not significantly different from +RPMI-8226 vs +RPMI-8226+sBCMA
  4. Please add the BCMA expression of KMS-11 multiple myeloma cell line. 
  5. In figure 4, the figure G is marked as F-bioluminescence images, please correct it. 
  6. Please add the statistics in supplementary figure 3.
  7. Did the authors have any evidence/any sample collection? from CYAD-211 clinical trial that the CAR T cells are exhausted at the time of disease relapse? It will contribute to the data for lack of persistence. 
  8. As authors commented it is likely that the additional engineering is needed to precent host versus graft reaction. 

Reviewer 2 Report

Comments and Suggestions for Authors

I have no major criticisms of the paper: I think it represents an intriguing approach into bringing allogeneic T cells into the fold re: engineering for adoptive T cell therapies. 

I would recommend that the authors make their figures a bit clearer: using larger dots for their graphs would be helpful. Further, if the y-axis of the graph in fig 1G was set at -0.5 or -1, it would make the unresponsiveness of the modified T cells "pop" more, so to speak. 

Barring that, great job, everyone! 

Reviewer 3 Report

Comments and Suggestions for Authors

The authors here developed a strategy of miRNA-based shRNA targeting CD3ζ to reduce the level of TCR components in allogeneic CAR T-cells. The in vitro and in vivo experiments are fully performed to validate the feasibility of this kind of method in decreasing graft-versus-host disease (GvHD) of allogeneic CAR T-cells. The results were quite encouraging. Some points need to be noted before the final acceptance for publication.

1. In the introduction part, the description of miRNA-based shRNA and their applications is still not detailed. For instance, what kinds of miRNAs (miR30a, miR155, miR451, etc) have been engineered for gene silencing? How is going in the applications of CAR-T cell engineering?

2. The whole gene sequence of miRNA-based shRNA (tCD19-CD3ζ miRNA-shRNA constructs and BCMA-specific CAR-CD3ζ miRNA-shRNA constructs) in this study can be also provided in the supplementary material.

3. There is seemingly a mistake in Figure 1E and 1F. The gray column should be Mock T-cells shown as the white. Please check again.

4. In the discussion part, the comparison of miRNA-based shRNA gene silencing and CRISPR-based gene knockout can be shown with more details, focusing on the respective advantages and limitations.
